# Rural Community Engagement for Health Disparities Research: The Unique Role of Historically Black Colleges and Universities (HBCUs)

**DOI:** 10.3390/ijerph18010064

**Published:** 2020-12-23

**Authors:** Lorraine C. Taylor, Charity S. Watkins, Hannah Chesterton, K. Sean Kimbro, Ruby Gerald

**Affiliations:** 1Juvenile Justice Institute, North Carolina Central University, Durham, NC 27707, USA; 2Department of Social Work, North Carolina Central University, Durham, NC 27707, USA; cwatkins33@nccu.edu; 3Justice, Law, and Criminology Program, American University, Washington, DC 20016, USA; hc7869a@student.american.edu; 4Biological and Biomedical Sciences, North Carolina Central University, Durham, NC 27707, USA; kkimbro@nccu.edu; 5Community Liaison, North Carolina Central University, Durham, NC 27707, USA; ruby.gerald@gmail.com

**Keywords:** HBCUs, rural, minority health disparities, community engagement

## Abstract

Reducing health disparities in rural communities of color remains a national concern. Efforts to reduce health disparities often center on community engagement, which is historically the strategy used to provide rural minority populations with support to access and utilize health information and services. Historically Black Colleges and Universities (HBCUs), with their origins derived from social injustices and discrimination, are uniquely positioned to conduct this type of engagement. We present the “Research with Care” project, a long-standing positive working relationship between North Carolina Central University (NCCU) and rural Halifax County, North Carolina, demonstrating an effective campus–community partnership. The importance of readiness to implement Community-based Participatory Research (CBPR) principles is underscored. As demonstrated by the NCCU–Halifax partnership, we recommend leveraging the positive associations of the HBCU brand identity as a method of building and sustaining meaningful relationships with rural Black communities. This underscores the role and value of HBCUs in the health disparities research arena and should be communicated and embraced.

## 1. Introduction

Research Centers for Minority Institutions (RCMI) grants were created to advance the science of minority health and health disparities through development and implementation of collaborative solutions to improve minority health outcomes and eliminate health disparities. Despite efforts to reduce health disparities, long standing social, environmental, and economic challenges to rural healthcare remain a national concern. In addition to geographic expansiveness and limited availability of health care options, barriers to rural health care also include socio-economic factors such as poverty, lack of transportation, and lack of health insurance [1]. Barriers to access and use of healthcare disproportionately impact rural communities [1,2]. Historical discrimination against Black patients in research settings has instilled a lack of trust in institutions and formal health care services among many rural Black communities, which also contributes to disparities in health [3]. This explains years of data indicating the increased rates of disease and poor health outcomes for rural Black populations—trends that are manifesting in the current COVID-19 pandemic [4,5]. Individual health risks such as higher rates of smoking and obesity also adversely affect rural health outcomes [6,7]. These health disparities reflect the confluence of factors that drive health disparities for rural minority populations.

Efforts to reduce health disparities often center on community engagement, which is a strategy used to provide rural minority populations with support to access and utilize health care services. Effective community engagement with rural minority populations has been subject to discourse, and best practices have been proffered by researchers. Eliminating health disparities often involves addressing barriers to access and use of health care services, work that is exacerbated by racial tensions, lack of trust, and disconnection between rural minority populations, and the health care system. In the southern United States, historically Black colleges and universities (HBCUs) have been actively involved in health disparity reduction using community engagement strategies with rural minority populations [8,9]. Given their histories and explicit emphasis of social justice as part of their missions, HBCUs are uniquely positioned for successful community engagement with rural minority populations [10]. In this paper, the potential successes and challenges of HBCU-led health disparities research in rural communities are discussed. Findings from an HBCU-led community partnership in one rural county in North Carolina are presented, with insights that can be used to promote more effective solutions for eliminating health disparities among rural minority populations.

Understanding connections between academic institutions and communities is a natural starting point for examining rural minority health disparities. Much has been published about the nature of university–community relationships, including the use of this model for addressing health concerns in various minority populations [11], and ways in which HBCUs are uniquely positioned for community engagement with minority, low-income, and underserved populations [12,13]. Civic engagement is an integral part of the mission of many HBCUs; this can be leveraged by health disparity researchers to build relationships with local communities [10]. This is particularly true of public HBCUs that have historically boasted institutional commitment to community service and empowerment [10]. Academic–community partnerships can play an important role in improving health outcomes when built on trust and around the specific needs of the community [14], once again underscoring the unique opportunities of HBCUs. Collaborations between academic researchers and community organizations have great potential for improving quality of life in rural areas, particularly when community stakeholders are actively involved in research planning and implementation [15].

The Community Based Participatory Research (CBPR) framework is the gold standard for approaching community-based research. Israel et al. [16] outlined the nine Principles of CBPR, which include building on community strengths; dissemination of findings and knowledge gained to all partners; and establishing a long-term commitment to the process. This framework has been implemented in health disparities research projects with rural populations. For example, Glover & Xirasagar et al. (2009) [13] incorporate a CBPR approach into the training of health professionals in an HBCU located in North Carolina, and Gragg and Mitchell et al. (2015) [17] describe findings from a CBPR-based program focused on enhancing health equity in local communities in a Southern state. Given the dedication to social justice and community engagement that often characterizes HBCUs, university-community partnerships based on the CBPR framework are especially likely to be impactful. Snydor et al. (2010) [18] describe the “fit” between HBCUs and CBPR, underscoring the utility of this framework for conducting research on a variety of topics of interest to minority populations. Next, we present an example of an HBCU-community partnership for health disparities research. Although full implementation of CBPR has been associated with positive health-related outcomes for minority populations [8,19,20,21,22,23], understanding how to establish research partnerships between HBCUs and communities requires attention. HBCUs must determine their “readiness” for CBPR implementation when approaching health disparities research with rural minority populations. Although HBCUs are uniquely positioned for this work, they also face a unique set of research challenges that must be addressed for HBCU-community research partnerships to be impactful and sustainable [18,24,25]. A description of a project that highlights the readiness for CBPR notion and identifies strategies for overcoming challenges follows.

## 2. Materials and Methods

### 2.1. The “Research with Care” Project and CBPR Readiness

An HBCU–community partnership for health disparities research between a mid-sized four-year university and a rural community in a county Southern state was formed. With a population of about 50,000 people, Halifax County, NC, spans over 700 square miles. From 2010 to 2019, the population decreased by 8.4%. In 2019, 54% of the population was Black, 40% White, 4% American Indian, and 3% Latino or Hispanic [26]. The poverty rate is 26%. The county unemployment rate in October, 2020 was 56% higher than one year before, reflecting the effect of the COVID-19 pandemic [27]. Health disparities such as increased rates of diabetes, cancers, and obesity are found for the Black population in this county, which remains largely segregated along racial lines. The university–community partnership originated as part of a research project focused on diabetes in the county’s Black families. This diabetes project was instrumental in the university being awarded a U54 (Research Centers for Minority Institutions, RCMI) research grant, to further study health disparities in rural Halifax County. Included in the RCMI grant was a community engagement core (CEC) that enabled faculty researchers at the university to connect with populations of interest in rural Halifax County. “Research with Care” was selected as the CEC project name, a title that embodied the true spirit of the collaboration between university and the Halifax County community.

In order to build trust and establish positive working relationships between the CEC and the Halifax County community, the Research With Care project emphasized several important themes, as described by Snydor et al. (2010) [18]. These themes require operationalization for HBCU–community collaborations to be established and sustained. Converting these themes into guiding principles defines the “readiness” work necessary for the implementation of CBPR. How did the Research with Care project actualize these CBPR readiness themes? Next is a brief description of how this occurred.

First is the notion that communities are agents, not objects. Because the perspectives, needs, and recommendations from Halifax County were germane to the success of this project; in addition, because none of the university researchers hailed from Halifax community, it was important to include a community liaison as part of the research team. To this end, a part-time position for a community liaison was included as part of the CEC grant and filled by a resident of Halifax County who was born and raised in the community and was well-connected within the county and region. The community liaison was central in all aspects of the Research with Care project, from making introductions with key community stakeholders to attending national conferences as part of the project team. Having a community liaison with a deep understanding of the history, economy, culture, and health needs of the area was critical for the success of the project. This person facilitated recruitment of participants and ensured that community members understood the goals of the project. Consistent with the idea that communities are not to be acted upon, but rather, collaborated with, hiring a community liaison was an important step in preparation for the research partnership focusing on health disparities.

The second aspect of CBPR readiness is the notion that partnerships are long-term relationships. A criticism of university-community partnerships is that they are often limited in duration because sustaining the relationships becomes difficult once funding ceases [28]. Although the end of funding may be inevitable, creative ways to sustain relationships over time must be considered. For example, faculty researchers can communicate regularly with community partners to share updates on project outcomes and news. Phone calls, newsletters, online postings, or other originative methods can be considered. Maintaining communication with Halifax County partners upon completion of the original diabetes study set the stage for the community engagement plan that was included in the RCMI grant application. Once awarded, the new RCMI-CEC was able to hit the ground running because of positive relationships that were maintained over time with the Halifax County partners.

Third is the notion that communities are receivers and providers of new knowledge. In addition to scientific research, for community engagement efforts to be successful, researchers must consider what new knowledge may be gained through partnerships with community members. Bi-directional relationships create positive collaborations that are beneficial to all [29]. Rural health disparities research may be conducted with the goal of helping people in these contexts to overcome problems and to promote health and well-being. Well-intentioned efforts may fall short, however, if researchers fail to understand the perspective of the population of interest and the context in which their research will take place. To better understand the rural context, project team members from the university embark on an annual tour of Halifax County, which is led by the community liaison. These driving tours connect university researchers with the Halifax community. The Halifax-long tours include stops to meet with community members in order to discuss ideas and plans for health disparities research. The tours also provide a sense of the geography of the vast and largely agricultural county, as well as providing the chance for team members to see schools, churches, health care facilities, prisons, and other institutions throughout the county. Dialoguing with community members during these visits is an invaluable part of building the relationships that facilitate the implementation of the CBPR model.

The final aspect of CBPR readiness is that partnerships should be mutually beneficial. Again, the notion of bi-directional relationships must be considered. What will the research team “give back” to the community that will be participating in research studies? Beyond individual research incentives for participation, researchers must consider how the community benefits from the collaboration. Careful thought about this in advance of embarking on data collection will support the relationship with community partners and facilitate accomplishing the research goals. In addition to conducting the various health disparity research projects in Halifax County, the CEC included several activities that were designed to “give back” to the community. For example, it was requested that the researchers share information on specific health topics. A monthly newsletter on a health topic was produced by the CEC and distributed at local churches. These reader-friendly and informative newsletters were well-received by community members. The CEC also conducted several health fairs in the county, providing services such as cholesterol screenings, hearing checks, and blood pressure monitoring. Health fairs provided community members with a fun and relevant opportunity to receive basic health care and information, while providing researchers with an opportunity to recruit participants for studies and share information about/on various health disparity topics.

This particular CEC is an example of a CBPR-ready effort, based on its adherence to the readiness criteria outlined above. As an HBCU, the university was able to build a recognizable brand identity with “Research with Care” in the community, and this brand was positively associated with the HBCU. The “Research with Care” logo was featured prominently on project materials and incentives, and was displayed at all community events. This brand, along with the many steps taken to prepare for effective collaboration as defined in the CBPR model, enabled health disparities work to flourish in Halifax County. For example, the Research with Care project sponsored several community health fairs for the local population (giving back to the community) and also facilitated data collection for NCCU faculty research projects. These are examples of the success of the collaboration. Although this particular collaboration was CBPR-ready, a number of important challenges emerged that highlight potential pitfalls for research through HBCU-community partnerships. Identifying these challenges and the strategies implemented to overcome these challenges may be especially helpful for promoting effective partnerships for future health disparities research between other HBCUs and community partners.

### 2.2. Rural Research Challenges

On several levels, conducting research in rural communities presents challenges for HBCU researchers. At a basic level, the travel distance between rural communities and HBCUs located in urban centers may hinder the research process. HBCUs located in more rural areas may not have to travel great distances for community-based research, but transportation issues may be still a concern. Recruitment of research participants, data collection, and dissemination of research findings is hampered when the researchers cannot easily reach the community, because of distance or transportation issues. Prior to conducting a research study, researchers must establish rapport, convey researchers’ intentions, and obtain community buy-in, which is often achieved through face-to-face interactions over time. This need for community engagement as a precursor to community research thus requires HBCU researchers to travel back and forth across long distances to reach areas that are often remote, and this can place a strain on researchers’ financial resources, time, and commitment to the project. The distance between HBCUs and rural communities is further complicated by the lack of technological infrastructure within rural communities. Compared to residents in urban areas, residents in rural communities are significantly less likely to own a home computer (laptop or desktop) or a smartphone [30]. As a result, rural residents are also significantly less likely to have access to high-speed internet (Martin, 2018; Ryan, 2018) [30,31]. As the dependence on internet access for research recruitment efforts and data collection methods continues to grow, researchers seeking to engage rural communities may struggle to identify potential participants and to efficiently administer survey instruments. Although there are no easy solutions to address the distance challenge, careful planning and open communication with community partners before, during, and following the research process is important to maintain. Creative solutions that leverage available assets can also help, such as the use of telecommunication and possible in-kind supports through the university for transportation needs.

### 2.3. The Cost of Collaboration

HBCUs and Predominately White Institutions (PWIs) sometimes work together in order to conduct health disparities research with minority populations. Although such collaborations may bring needed resources and research infrastructure to HBCUs, these collaborations present challenges that may overshadow gains derived from working together. For these HBCU-PWI partnerships to be successful, they must be “built on mutual respect, mutual trust, and mutual benefit,” [32]. The legacy of racism, one of the sparks that led to the creation of HBCUs, and the resulting mistrust, suspicion, and exploitation must be addressed. The legacy of racism must be considered in relation to role assignment, decision-making processes, data ownership, and dissemination of information. When HBCUs have established trusting relationships with rural communities of color, collaboration with PWIs may be met with skepticism by community members [33].

Several problematic aspects of HBCU-PWI collaborations are identified in a study of such partnerships, including the fickle nature of the partnership; lack of transparency and clarity in communications; and PWI claims of ownership for ideas proposed by the HBCU [34]. This study also highlights benefits to such collaborations, but steps should be taken to prevent problems that may taint cross-institution relationships and negatively impact the community. Researchers should model exemplary HBCU-PWI collaborations, such as the Meharry-Vanderbilt Alliance, the Georgetown University and Howard University Center of Excellence for Health Disparities, and the South Carolina Clinical and Translational Research Institute’s partnerships with Clemson University and South Carolina State University, all of which have addressed the health-related needs of underserved, uninsured communities [23,35]. 

### 2.4. Information Sharing

A major challenge in the optimization of academic-community partnerships is the lack of information sharing. Collaboration with the non-academic world is an important aspect of university research, and in order for community engagement to flourish, information must be shared among all stakeholders [36]. The sharing of research findings with the communities from which data were extracted is important both for the researchers and for the community itself. For the researchers, sharing study results promotes community trust. In addition to building positive rapport through the sharing of findings, participants are also able to review data and their interpretation, which may be especially prior to publication of the research findings [37]. Validation of study findings by participants provides credibility to the researcher within the community, potentially facilitating the identification of community members who are willing to participate in future projects led by the researcher.

The lack of information sharing with community collaborators may occur for several reasons. Despite the investment of time, money, and energy to collect data on the front end of the research process, HBCU researchers, especially those at public, state-sponsored institutions, may not have the resources necessary to process, analyze, and summarize findings from the data analyses in a timely manner. The lack of financial resources at many HBCUs is a serious challenge for researchers [38]. Examples of research-related resources needed but potentially lacking for HBCU researchers include access to statistical software to prepare and analyze quantitative datasets; training in research methodology; assistance with data analysis; and a lack of funding to address those barriers [25]. In addition to resource challenges, HBCU researchers may also lack the time necessary to write up research results. With higher course loads and more administrative responsibilities compared to researchers at PWIs, HBCU researchers may find it difficult to dedicate time to what may be an unfunded component of a research project [25]. Despite these challenges, HBCU researchers should include information sharing strategies as part of the research plan in the initial interactions with community partners. Asking partners about preferred methods for receiving information will provide researchers with concrete measures that can be integrated into the research plan. Emphasizing a bidirectional relationship, including strategies for sharing research findings, actualizes the tenets of CBPR readiness pertaining to the mutual benefits of the research collaboration and is an important aspect of promoting long-term positive relationships between HBCUs and their community partners [29,37].

## 3. Conclusions

The Research with Care project, a successful collaboration between an HBCU and a rural community in North Carolina, illustrates how a meaningful, mutually beneficial research partnership can be forged when communities are viewed and treated as active collaborators with valuable knowledge to contribute. By focusing on CBPR “readiness,” collaboration pitfalls can be avoided. Given the ongoing challenges of addressing health disparities among underserved, rural populations, HBCUs are resources that should continue to be leveraged. Although a number of important considerations are presented in this paper, a few limitations must be acknowledged. This paper presents only one example of a HBCU–rural community partnership. There are facets of this partnership that may be unique and therefore not directly applicable to other academic-community partnerships. The Research with Care project was based on a community-based diabetes education and prevention program for rural African Americans. HBCU-community partnerships that are aimed at addressing other health disparity issues or focusing on other populations of interest may require consideration of different outreach and implementation strategies appropriate for use with the populations of interest. Variations in rural contexts must also be considered. Halifax County may differ from other rural counties in terms of proximity to urban centers, technological resources, and existing relationships or connections to HBCUs, all of which may have indirectly facilitated the development of the academic-community partnership between Halifax County and NCCU.

In the ongoing effort to eliminate health disparities in rural underserved communities, the current sociopolitical climate and global health pandemic should not be overlooked. Backlash against African Americans striving for fair treatment and justice currently plagues the nation, reminding many rural residents in southern United States of pre-Civil Rights era realities. Data show the impact of the COVID-19 global pandemic on rural minority communities to be especially negative. As of July 2020, 96% of all rural counties had at least one confirmed case of COVID-19 and half of all rural counties had experienced at least one COVID-related death [39]. It should also be noted that among the top COVID-19 hotspots—or counties with the highest number of recent cases per resident—an overwhelming number are located in nonmetropolitan areas [40]. Several factors may place rural communities at risk for growing COVID-19 cases including rural residents typically being older with underlying health conditions, uninsured due to health insurance costs, and living in a medically underserved community [40]. Partnerships between HBCUs and rural communities for health disparities research remain relevant to the southern US and are likely to be needed in years to come as the impacts of these threats to the health and well-being of minority populations continue to emerge.

The need for HBCUs to be active in rural community engagement is clear. Health disparities research continues to thrive at HBCUs through mechanisms such as the Research Centers for Minority Institutions (RCMI), with examples of successful projects from around the nation [41]. By preparing for implementation of CBPR principles, leveraging the positive associations of the HBCU brand identity, and avoiding common pitfalls, HBCU researchers can build and sustain meaningful relationships with communities. Communicating the success of these partnerships to the public is essential and the “value added” by HBCUs in the health disparities research arena should be underscored [38].

Despite years of focus and a proliferation of research studies, health disparities remain and continue to disadvantage rural minority populations. Rural community engagement efforts by HBCUs may be an especially effective strategy for combating these disparities, given their unique histories and missions related to marginalized communities. It remains to be seen if the trajectory of health disparities among populations of color changes for the better in the coming years, but researchers from HBCUs are currently poised to develop and sustain positive relationships with minority communities in order to help move the needle in the right direction.

## Data Availability

No new data were created or analyzed in this study. Data sharing is not applicable for this article.

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
