# Peer review of "Rural Community Engagement for Health Disparities Research: The Unique Role of Historically Black Colleges and Universities (HBCUs)"

_ijerph, 2020, doi:10.3390/ijerph18010064_

Round 1

Reviewer 1 Report

I found this a difficult paper to review as it is not reporting the results of a study, but rather discusses one specific aspect of a research project, namely CBPR. It reads more as a paper about research challenges in CBPR based on a 'case' study. It may be that it would be more useful to present it as such. 

The premise of the paper is about HBCUs. This is to my knowledge, is an American phenomenon – how might these findings apply to other settings outside of the US? In a similar vein, 'HBCU' is an  American phrase so I would recommend writing it in full in the title.

The paper mentions that there are several evidence-based frameworks that can be employed in community engagement, but fails to identify or discuss these. What are they? Which did you choose to use? How did they support the implementation of the project. Beyond the readiness themes, the paper mentions frameworks but does not  engage with them. 

There could be more  emphasis on the outcomes of the project  (in relation to health disparities). Might this provide some insight into the success (or otherwise of the community engagement? Would it be possible to identify what worked well and what worked less well? Lessons learned?

Quite a substantial number of the references are 2010 or earlier. Is there more recent work in the field that you can draw on?

Reviewer 2 Report

This is an informative paper with good potential for contributing to the literature on Historically Black Colleges and Universities (HBCU) and the community engagement mission of such institutions. I have a number of comments for the authors to consider.

The term, “successful” (or “success” / “successfully”), is used throughout the paper, but “success” is not substantively defined. The authors should explicitly state their operational definition of a successful community engagement program. By operational definition, I mean specific, measurable outcomes that lead to quantitative indicators of success / failure. These days in U.S. academia, the term, “success,” is too often ill-defined and over-used.

Is there any hard evidence that HBCU-led community engagement efforts have actually reduced black-white health disparities with respect to diabetes, hypertension, or any illness? The authors mention, for example, that HBCU-PWI (Predominantly White Institutions) partnerships have been “successful” (line 231); however, the authors are not specific about what they mean by success.  Even some anecdotal (i.e., qualitative) evidence would be helpful.

The authors might address the issue that the rural black population is small and declining, even in the South, and that the rural black population, similar to its white counterpart, is disproportionately elderly.

Regarding population composition, the authors break down the Halifax county population (lines 100-101) but do not mention Latinos. What percentage of the county population is Latino? I ask because it is well-known that North Carolina has seen a tremendous influx of Latinos, as have other southern states in the U.S.

The authors mention that HBCUs are “overwhelmingly located in urban centers” (line 192), yet there are many HBCUs in rural or non-metropolitan areas. For example, in Mississippi, two of the three state-sponsored HBCUs (Mississippi Valley State University and Alcorn State University) are in relatively rural areas. The third one is in the state capital (Jackson State University). Of course, one of the most famous HBCUs, Tuskegee University, is outside of an urbanized area. The authors might give us some statistics on the urban-rural breakdown of HBCUs, most but not all of which are in the South.

The authors might distinguish between state-sponsored and privately funded HBCUs in discussing institutional advantages and disadvantages. My impression is that the state-sponsored HBCUs have historically been service-oriented institutions dedicated to serving disadvantaged black populations, whereas the privately funded HBCUs were more oriented to educating the black white collar and professional classes. It might be argued that state-sponsored HBCUs have a legacy advantage when it comes to service and community engagement projects.

Finally, the manuscript needs more copy-editing and the writing could be improved so that the text is more readable and concise. Overall, the manuscript is clearly written, but the writing is sorely in need of polishing. For example, the writing is somewhat repetitious, and Table 1 (lines 119-120) is unnecessary.

In conclusion, this paper is informative. However, the authors need to explicitly and substantively define “success” and to present evidence (qualitative or quantitative) that HBCU community engagement actually reduces racial disparities in health outcomes. The authors also need to more carefully consider the diversity of HBCUs (rural / urban, public / private) when discussing the unique role of these institutions in rural community engagement as it relates to reducing racial disparities in health outcomes.

Round 2

Reviewer 1 Report

Accept 

Author Response

On behalf of my co-authors, I thank you for your time and comments.  

Reviewer 2 Report

Lines 7-10 The affiliations numbered 3-5 are not assigned to any authors.

Line 21 Spell out CBPR first and then abbreviate.

Line 23 “This”... what? Antecedent is unclear. Is it “partnership”?

Line 57 Emphasize the potential successes.

Lines 84-84 The sentence should read, “...findings from a CBPR-based program...” Insert “a”

Line 121 Delete reference to Table 1.

Line 222 “...infrastructure that is especially evident...” This sentence is unclear.

Lines 225-226 True, “racism...sparked the creation of HBCUs...” But also keep in mind that these institutions and other schools were built by a southern black leadership class that emphasized self-help and group solidarity as a response to racism. See John Sibley Butler's book on black entrepreneurship and self-help (SUNY Press).

Lines 236-239 The “exemplary” HBCU-PWI partnerships listed are all urban-based institutions. Are there examples of HBCU-PWI partnerships for rural / non-metropolitan areas?

Line 241 Where is the article “forthcoming”? Provide the journal / book title (lines 458-459).

Line 304 “...relevant to the southern US...” Insert “the”

Line 308 “...continues to thrive...” A supporting reference is needed for this statement.
